# Role of informal healthcare providers in tuberculosis care in low- and middle-income countries: A systematic scoping review

**Poshan Thapa**[1]*, **Rohan Jayasuriya**[1], **John J. Hall**[1], **Kristen Beek**[1], **Parthasarathi Mukherjee**[2], **Nachiket Gudi**[3], **Padmanesan Narasimhan**[1]

**1** School of Population Health, University of New South Wales, Sydney, Australia, **2** Liver Foundation, Kolkata, West Bengal, India, **3** The George Institute for Global Health, New Delhi, India

* thapaposhan2009@gmail.com

**Data Availability Statement:** All relevant data are within the paper and its Supporting Information files.

## Abstract

Achieving targets set in the End TB Strategy is still a distant goal for many Low- and Middle-Income Countries (LMICs). The importance of strengthening public-private partnership by engaging all identified providers in Tuberculosis (TB) care has long been advocated in global TB policies and strategies. However, Informal Healthcare Providers (IPs) are not yet prioritised and engaged in National Tuberculosis Programs (NTPs) globally. There exists a substantial body of evidence that confirms an important contribution of IPs in TB care. A systematic understanding of their role is necessary to ascertain their potential in improving TB care in LMICs. The purpose of this review is to scope the role of IPs in TB care. The scoping review was guided by a framework developed by the Joanna Briggs Institute. An electronic search of literature was conducted in MEDLINE, EMBASE, SCOPUS, Global Health, CINAHL, and Web of Science. Of a total 5234 records identified and retrieved, 92 full-text articles were screened, of which 13 were included in the final review. An increasing trend was observed in publication over time, with most published between 2010–2019. In 60% of the articles, NTPs were mentioned as a collaborator in the study. For detection and diagnosis, IPs were primarily involved in identifying and referring patients. Administering DOT (Directly Observed Treatment) to the patient was the major task assigned to IPs for treatment and support. There is a paucity of evidence on prevention, as only one study involved IPs to perform this role. Traditional health providers were the most commonly featured, but there was not much variation in the role by provider type. All studies reported a positive role of IPs in improving TB care outcomes. This review demonstrates that IPs can be successfully engaged in various roles in TB care with appropriate support and training. Their contribution can support countries to achieve their national and global targets if prioritized in National TB Programs.

**Funding:** The author(s) received no specific funding for this work.

**Competing interests:** The authors have declared that no competing interests exist.

## Introduction

Tuberculosis (TB) was declared a global public health emergency in 1993, and remains one of the leading causes of death worldwide [1, 2]. In 2019, there were an estimated 10 million new cases of TB and 1.2 million deaths among HIV negative people, with an additional 208,000 deaths among HIV positive people [3]. Worldwide TB causes significant loss of productivity, catastrophic health expenditure, and is the 11th leading cause of years of life lost [4, 5]. Between 2015–2019, the global cumulative reduction in incidence and mortality was 9% and 14%, respectively. Despite having a progressive reduction, countries with a high TB burden were far from achieving the 2020 milestone of reducing incidence by 20% and mortality by 35% [3, 6].

Mandatory TB case notification is an integral component of the World Health Organization (WHO) End TB Strategy [7]. There was a noticeable increase in global tuberculosis case notification in 2018, but with a large gap of 3 million missing cases [3]. Seven Low- and Middle-Income Countries (LMICs) with a dominant private health sector account for almost two-thirds of the world's missing TB cases. This gap between estimated and reported cases is due to the under-reporting of detected cases as well as the under-diagnosis of new cases, partially due to the significant private sector presence in health system provision [3, 8]. The significant contribution of the private sector in tuberculosis care is evident from patient pathway analysis [9, 10] and national TB drug sale data [11]. While private sector engagement has been endorsed in global TB strategies since the 1990s, there are ongoing challenges for National Tuberculosis Programs (NTPs) in many countries. Reasons for this include heterogeneity of the private workforce, undocumented size and scope of practice, the lack of legislative frameworks, inadequate health information system, and problems with role clarity [2, 12].

To achieve national and global TB targets, it is essential to engage all health providers, public and private, as well as formal and informal, in line with the WHO Public-Private Mix (PPM) strategy to ensure every case of TB is appropriately detected and receives timely treatment [8, 13]. The Lancet Commission on TB has recommended prioritizing engagement with the private sector in TB programs [2], and this is particularly important for LMICs where the private sector is a predominant provider in health systems [12, 14]. Within the private sector, a large cadre of providers operate outside the formal health system whilst lacking appropriate qualifications for the health services they deliver [15]. In the current literature, commonly used terms to describe this health workforce are Informal Providers and Informal Healthcare Providers (IPs) [16–19]. The term encompasses a wide variety of providers who are broadly distinguished by the nature of their practice and include village doctors, traditional healers, drug compounders, traditional birth attendants, and untrained allopathic practitioners [16, 20]. Although unrecognized and undocumented in many settings, a considerable amount of evidence establishing their contribution to the health system exists. This is especially evident in rural and underserved areas of LMICs, where primary care-seeking from IPs by patients ranges from 65%-90% [16, 19, 21]. The pivotal role of IPs in TB care is increasingly recognized at the global level and evident through multiple patient health-seeking behaviour studies [22–26], as well as through some large scale surveys [27, 28]. However, their prioritization as an important provider in TB care remains the missing piece in NTPs of many LMICs.

Non-engagement of all providers in TB care can potentially limit the success of NTPs. Consequences can include increased community transmission, delayed diagnosis and treatment, drug resistance, catastrophic health expenditure for patients, and impaired tuberculosis monitoring and evaluation systems [8, 23, 29]. The role of different providers varies by qualifications, national regulations, and their practice; however, it is crucial to recognize and engage all providers. The inability to integrate IPs not only represents a missed opportunity but has also limited our understanding of their role in TB care. A substantial body of research suggests that

they are the first point of contact for a significant proportion of TB patients [27, 28], but beyond that point, our understanding of their role in the prevention, diagnosis, and treatment of TB is limited. Studies which have explored the role of IPs in tuberculosis care have found that they create community awareness, assist in the diagnosis of symptomatic cases, and conduct sputum collection [30–32]. However, to our knowledge, their role has not been systematically explored using an evidence synthesis approach. The previous reviews measuring the impact of a PPM model in TB care have primarily focused on the public and formal private systems. Information on IPs is cursory and does not provide clear insight into their role [33, 34]. A discussion paper published in 2011 attempted to explore the potential of engaging IPs in TB care, however, it lacks an appropriate search strategy and has a limited scope [35]. To meaningfully engage IPs in any NTP, the preliminary step must be to understand their current role. Therefore, in this paper, we have undertaken a systematic scoping review of the current literature to identify the role of IPs in TB care in LMICs.

## Materials and methods

A scoping review was identified as an appropriate method to achieve the research aim as it is commonly used to clarify working definitions, conceptual boundaries and to map the current status of an object or subject of interest within a particular field [36, 37]. The scoping review approach enables an understanding of the role of IPs in TB care in the available literature. This method is also relevant for disciplines with emerging evidence [38], with research on IPs being an evolving field. This scoping review was conducted using the framework developed by the Joanna Briggs Institute (JBI) based on the work of Arksey and O'Malley (2005) [39]. The framework consists of five sequential stages: 1) identifying the research question, 2) identifying relevant studies, 3) study selection. 4) charting the data, and 5) collating, summarizing, and reporting the results. Each stage is described in detail below. The scoping review is reported following the PRISMA extension for scoping reviews published in 2018, which is made available as a S1 File [40].

### Stage 1: Identifying the research question

All authors were involved in the discussion to finalize the research questions for this review. The research questions for the review were:

1. What are the roles of IPs in TB care?

2. What are the various types (classified based on their practice) of IPs engaged in these roles?

3. Which TB related outcomes are influenced by IPs role?

### Stage 2: Identifying relevant studies

The Inclusion criteria for the review was developed according to Population-Concept-Context (PCC) framework [39].

**Population.** As there is no uniform definition for IPs, we referred to previously published papers in this field [16, 17, 20, 21, 41, 42] and developed a summary table to identify the common elements in each definition, available as a S2 File. Hence, in this study, we define IPs as "individuals who are not affiliated or registered to any government body or institution, independently delivering some form of health services, and not possessing a recognized certification for the type of services they offer". The study did not include family or community members, caregivers, or volunteers. Similarly, Community Health Workers (CHWs) who

were formally trained and worked for an established institution (either private or public) were excluded from the review.

**Concept.** We defined role based on the functions performed by IPs in TB care. We used the WHO People-Centered Model of TB Care to classify functions. It provides a granular classification of TB functions and is best suited for this review to comprehensively outline the distinct role of IPs in TB care [43]. It first broadly divides TB care into three types; 1) prevention, 2) detection and diagnosis, and 3) treatment and support, and then further classifies 17 different functions under these three domains of care. The summary table of the model is available as a S3 File.

**Context and design.** The World Bank classification by income 2018–2019 was used as a reference to classify LMICs [44]. The review included articles using any study design if they reported any role of IPs in TB care in LMICs. Only studies published in English were included. There was no restriction on date considering the scarcity of evidence in this field. Exclusion criteria included review papers, editorials, commentaries, study protocols, conference abstracts, perspective pieces, or multiple papers from a single study.

**Search strategy.** A comprehensive search was carried out independently by two authors (PT and NG) in six electronic databases: MEDLINE, EMBASE, SCOPUS, Global Health, CINAHL, and Web of Science, between 15 June to 10 July 2019. The search was updated on 5 September 2020, but no new articles were retrieved. Four key terms were used for the literature search: "Tuberculosis" "Informal Healthcare Provider," Tuberculosis care," and "LMICs". The search strategy was developed in consultation with subject experts, including a qualified research librarian, and also by referring to previously published review studies on TB care and IPs [16, 20, 45]. A manual search of the reference lists of included studies was also undertaken. An example of a complete search strategy performed in MEDLINE (via Ovid) is available as a S4 File.

## Stage 3: Study selection

All identified studies from the electronic search were imported into EndNote X8, Clarivate Analytics, US [46]. After removing duplicates, in the first step, two authors (PT and NG) independently reviewed all the titles and abstracts. Any disagreement between these authors was resolved in discussion with a third author (RJ). As a second step, the full text of potential studies was reviewed by two authors (PT and NG). The reasons for excluding the full-text articles at the final stage of screening were recorded and are available as a S5 File. The final articles were discussed among all authors to rule out any disagreement through consensus. The study selection process is reported using the PRISMA-ScR flow diagram (Fig 1).

**Methodological quality assessment.** The goal of Quality Assessment (QA) is to provide an overview of the methodological rigour of the included studies. The findings from QA did not limit the inclusion of studies in the review. It was done independently by two researchers (PT, RJ), and any disagreement was resolved through discussion and consensus. We used Joanna Briggs Institute (JBI) standard critical appraisal tools for experimental and quasi-experimental studies [47]. Both these tools have a set of questions that help authors determine the methodological rigour of included studies. Each question is scored 'yes', 'no', 'unclear', or 'not applicable', but there is no overall quality score.

## Stage 4 & 5: Charting the data and collating, summarising, and reporting the results

Data were extracted using the predeveloped data extraction format in Microsoft Excel, which was tested with four articles and refined before the full extraction. Information retrieved

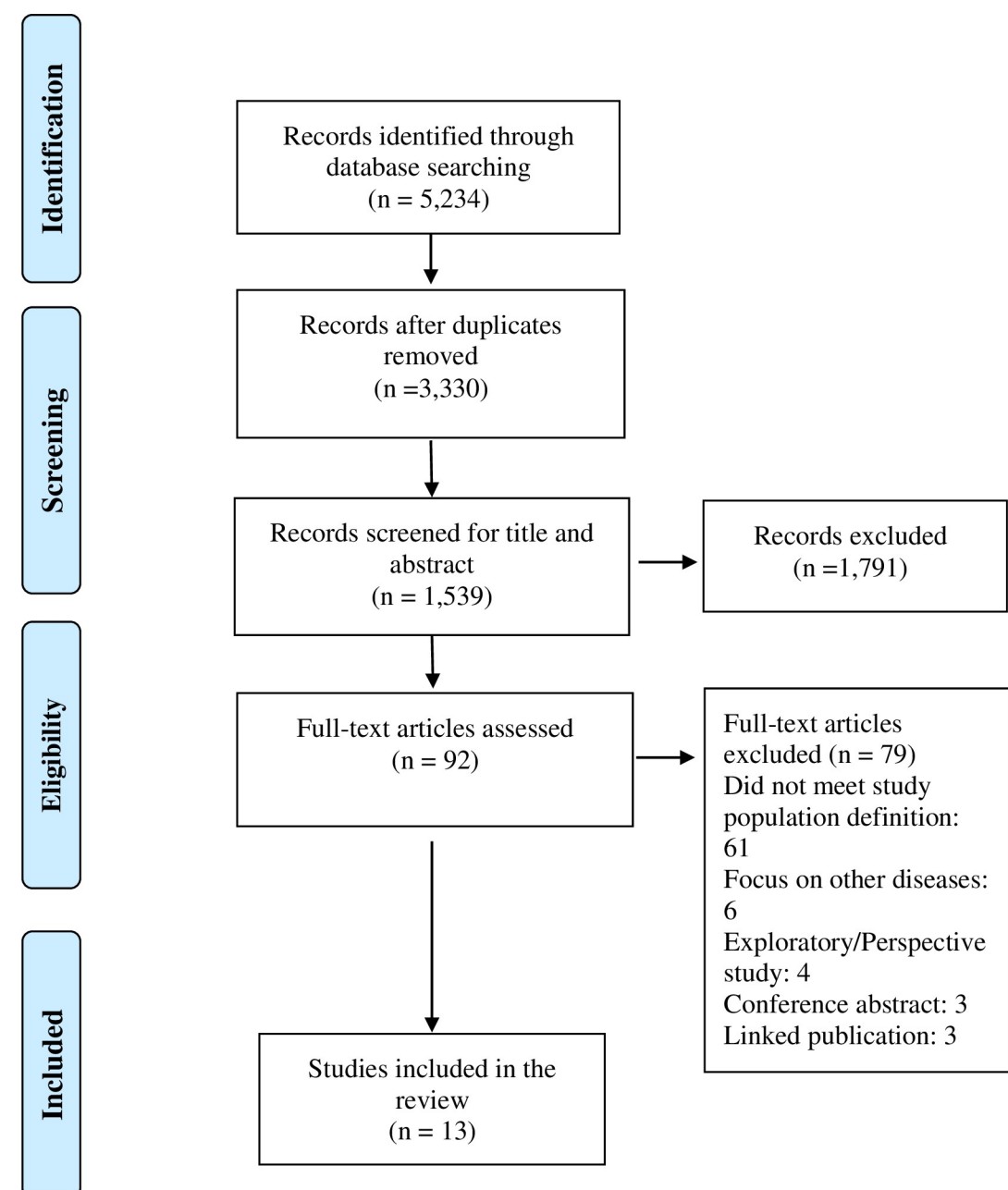

**Fig 1. PRISMA-ScR flow diagram of study selection and inclusion process.**

included study characteristics (year, objective, site, design, methodology), project characteristics (IP type, training, incentive, use of digital tools), a complete description of the role of IPs, study outcomes, and brief study findings. The goal of the scoping review is to provide an overview of the available literature, so all studies were included regardless of quality assessment outcome [39]. Information, including study objectives, project and IP characteristics, and study outcomes, are summarised descriptively. Data on the role of IPs were synthesized using the content analysis approach [38] based on the WHO People-Centered Model of TB Care.

## Results

### Characteristics of included articles

We retrieved a total of 5234 studies, of which 92 had their full text assessed.13 studies met the inclusion criteria and were included in the final review. "Incorrect population" was the major reason for excluding articles, meaning the participants did not qualify to be classified as IPs based on our study definition.

We noted an increase of published studies over time, as shown in Fig 2. About 50% (n = 6) of the studies were published between 2010–2019. Eight studies were from Africa and five from Asia. Based on Reichardt classification [48], a majority (n = 10) of the studies were quasi-experimental design (QED), and one study was a cluster randomised trial. We could not determine the design of the two studies as the information was limited [49, 50]. Most of the studies aimed to assess the feasibility, effectiveness, or acceptability of engaging IPs in TB care.

### Quality of studies

Out of 10 QED studies, only four studies scored more than 4 out of the 9 criteria listed in the checklist. The overall quality of the included studies was found to be low, especially for criteria 5, 6 and 9, indicating potential bias during measurement of outcomes pre and post [32, 51–57] and lack of appropriate statistical analysis [32, 51, 53–55, 57, 58]. Five studies reported having a control group (Q4), but in 3 studies, the information about the comparison group was not sufficient [54, 55, 57]. One cluster randomised study included in the review was of high quality (10/13) [31]. As it was a cluster randomised design, concealment and blinding of participants was not possible in the study. Future studies should employ standard research methods to address these identified methodological gaps. For detailed QA results, please refer to the S6 File.

### Characteristics of the providers and project in included studies

In more than 60% of the studies, NTPs were mentioned as one of the collaborators in the study [51–58]. None of the studies provided background characteristics of IPs, such as

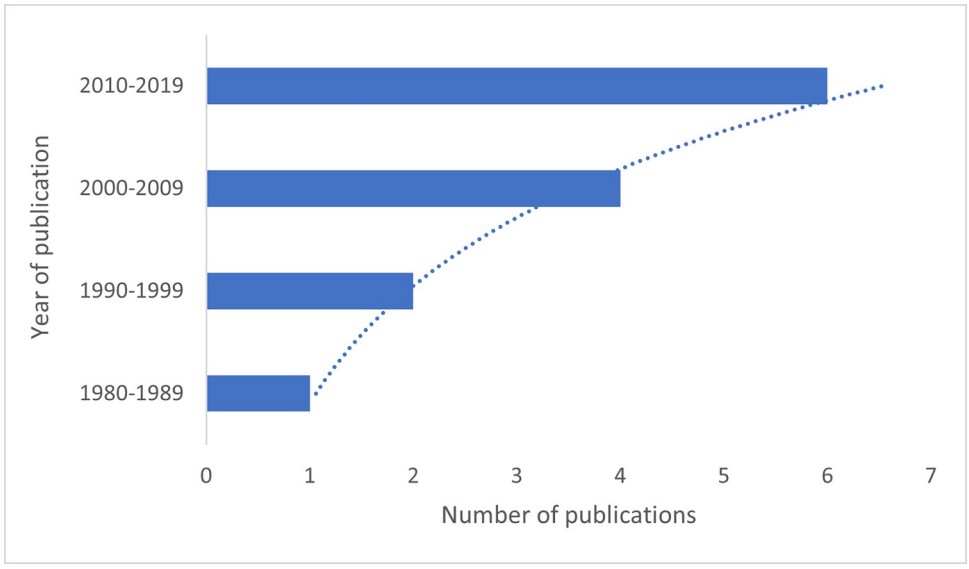

**Fig 2. Diagram showing studies based on year of publication.**

education, years of experience, primary training, system of practice etc. In our study, we classified IPs into four major types based on the nature of their practice, as shown in Table 1. The details of providers' classifications are available as a S7 File. The most commonly featured IPs in TB care were Traditional Health Practitioners (THPs) (n = 8) [30–32, 49, 51, 53, 55, 58], followed by traditional birth attendants (TBAs) (n = 3) [50, 56, 57] and drug sellers, storekeepers, and chemists (n = 3) [30, 31, 52]. The least featured IP type were untrained allopathic practitioners (n = 2) [30, 54]. In all studies, IPs were given some training in TB before their involvement, but the duration varied. Not all studies provided the duration of training, but in those that did, it was noted to be between 1–3 days [49, 52, 54–58], and in two studies, between 1–3 weeks [32, 51]. One study [53] provided details of monetary incentives to IPs. For all other studies, monetary incentives were not given for their work or information was not provided in the paper. None of the studies employed any kind of digital tool to support IPs to deliver assigned roles. Based on the nature of the roles assigned, patient referrals were made verbally or using a paper form [30, 32, 49, 51–54, 58] and similarly, information on treatment was documented in a logbook or patient treatment card [50, 54–57].

## Roles of IPs in TB care

In the majority of the studies, detection and diagnosis (n = 10) was the most commonly assigned role for IPs [30–32, 49–54, 58], followed by treatment and support (n = 6) [32, 50, 54–57]. In one study, they were assigned a prevention role, combined with detection and

**Table 1. Characteristics of included studies (n = 13).**

| Study | Country | Design | Provider types | Aim |
|---|---|---|---|---|
| Sima et al. (2019) [51] | Ethiopia | QED–one group posttest | Traditional Health Providers | To assess the role of traditional healers in the detection and referral of active TB cases. |
| Dutta et al. (2018) [30] | India | QED–prestest-posttest nonequivalent group | Chemist, Traditional Health Providers, Untrained allopathic practitioners | To identify TB patients emerging from under-reached communities through non-formal health providers. |
| Bello et al. (2017) [31] | Malawi | Cluster randomized trial | Storekeeper, Traditional health practitioners | To investigate the effectiveness of engaging informal providers to promote access in a rural district. |
| Colvin et al. (2014) [58] | Tanzania | QED–one group pretest-posttest | Traditional health practitioners | To evaluate the community-based project to improve TB case notification. |
| Kaboru et al. (2013) [53] | Burkina Faso | QED–one group posttest | Traditional health practitioners | To assess the contribution of traditional healthcare providers in TB case finding. |
| Simwaka et al. (2012) [52] | Malawi | QED–prestest-posttest nonequivalent group | Storekeeper | To determine the effectiveness and acceptability of a storekeeper-based referral system for TB suspects in urban settings. |
| Salim et al. (2006) [54] | Bangladesh | QED–posttest nonequivalent group | Untrained allopathic practitioners | To assess the feasibility and quality of village doctors' involvement in TB control program. |
| Harper et al. (2004) [32] | Gambia | QED–one group posttest | Traditional health practitioner | To evaluate the feasibility of involving traditional healers in tuberculosis diagnosis and treatment. |
| Colvin et al. (2003) [55] | South Africa | QED–posttest non equivalent group | Traditional health practitioner | To assess the acceptability and effectiveness of traditional healers as supervisors of tuberculosis treatment. |
| Kangangi et al. (2003) [56] | Kenya | QED–one group pretest-posttest | Traditional birth attendant | To evaluate the impact of district TB program performance of decentralizing TB treatment by providing ambulatory care in the hospital and peripheral health units and in the community. |
| Jagotal et al. (1997) [57] | India | QED–posttest nonequivalent group | Traditional birth attendant | To assess the utility of traditional birth attendants for supervised administration of anti-tuberculosis drugs to patients. |
| Balasubramanian et al. (1997) [50] | India | Unclear | Traditional birth attendant | To assess the feasibility of involving traditional birth attendants in case finding and door-delivery of drugs to tuberculosis patients. |
| Oswald et al. (1983) [49] | Nepal | Unclear | Traditional health practitioner | To explores if traditional medical practitioners can be used to reduce the communication gap between health posts and the community. |

diagnosis [31]. The details of their roles as classified by the WHO People-Centered Model of TB Care is presented in Table 2.

**Prevention.** In one study, IPs were assigned the role of conducting community awareness meetings where they provided information about the signs and symptoms of TB and the availability of free services at the local health facility [31].

**Detection and diagnosis.** Of 10 studies that focused on detection and diagnosis, in 90% of the studies, IPs passively identified patients who visited them at their place of practice and referred them to a nearby health facility if their symptoms resembled that of TB [30–32, 49, 51–54, 58]. In one study, IPs were assigned the role of active case finding. They were given a target population and tasked to identify TB suspects in the community [50]. In two studies, they collected sputum samples from TB suspects either at their place of practice or at the patient's home [30, 31].

**Treatment and support.** In all the studies with a treatment and support function (n = 6), IPs were assigned the role of administering DOT (Directly Observed Treatment) to confirmed TB patients. DOT was provided either at the IPs place of work or patient's home [32, 50, 54–57]. In none of the studies, IPs self-initiated the treatment of a tuberculosis patient. The TB medication was supplied to IPs by the government health center or NGO/ research institute based on the nature of the collaboration. In one study involving untrained allopathic practitioners, their assigned role was to refer patients to health centers if they had an adverse drug reaction [54].

## Roles of IPs in TB care by provider type

There was no major difference in role based on the type of provider (Fig 3). The most commonly featured providers in identified studies were traditional health providers, and they were assigned prevention (n = 1) [31], detection and diagnosis (n = 7) [30–32, 49, 51, 53, 58] and treatment and support (n = 2) [32, 55] roles. Traditional birth attendants were mostly engaged in treatment and support (n = 3) [50, 56, 57], followed by detection and diagnosis (n = 1) [50]. Drug sellers, storekeepers, and chemists were engaged in detection and diagnosis (n = 3) [30, 31, 52] and prevention (n = 1) [31]. Lastly, the untrained allopathic practitioners, the least featured provider type, were engaged in detection and diagnosis (n = 2) [30, 54] and treatment and support (n = 1) [54].

## Outcomes measured in the included studies

Outcome measures (Table 3) frequently used in studies identified for this review were treatment outcome (n = 6) [32, 50, 54–57] and referrals made (n = 5) [32, 49, 51, 53, 54] followed by case notification (n = 3) [30, 52, 58], and testing and treatment initiation rate (n = 1) [31]. There was an observed trend in outcomes measured in the studies; older studies have focused on treatment components and more recent studies on case identification and referral. All studies, including those found to provide high-quality evidence [30, 31, 52], have reported a positive role of IPs in improving TB care outcomes.

## Discussion

To our knowledge, this is the first review to systematically examine the role of IPs in TB care in LMICs. We identified that IPs were engaged in all three domains of TB care, predominantly in detection and diagnosis of disease, followed by treatment, and least with prevention activities. When we categorised IPs based on their practice, THPs were the most featured providers in TB care, and untrained allopathic practitioners were the least of the three categories. Interestingly, there was no variation in the role by provider type.

**Table 2. IPs role classified based on WHO People-Centered Model of TB Care.**

| Study | Prevention | | | Detection and Diagnosis | | | | Treatment and support | | | | | | |
|---|---|---|---|---|---|---|---|---|---|---|---|---|---|---|
| | Health promotion and education | Immunization | Latent TB infection services (3 functions) | Active case finding | Passive case finding referral | Clinical evaluation—TB | Lab—Sputum sample collection | Treatment Initiation | Treatment administration and observation | Monitoring treatment progress and response | Prevention and detection of adverse event | Diagnosis and treatment of adverse events | TB Lab monitoring | Counselling and support (2 functions) |
| Sima (2019) | | | | | ✓ | | | | | | | | | |
| Dutta (2018) | | | | | ✓ | | ✓ | | | | | | | |
| Bello (2017) | ✓ | | | | ✓ | | ✓ | | | | | | | |
| Colvin (2014) | | | | | ✓ | | | | | | | | | |
| Kaboru (2013) | | | | | ✓ | | | | | | | | | |
| Simwaka (2012) | | | | | ✓ | | | | | | | | | |
| Salim (2006) | | | | | ✓ | | | | ✓ | | ✓ | | | |
| Harper (2004) | | | | | ✓ | | | | ✓ | | | | | |
| Colvin (2003) | | | | | | | | | ✓ | | | | | |
| Kangangi (2003) | | | | | | | | | ✓ | | | | | |
| Jagotal (1997) | | | | ✓ | | | | | ✓ | | | | | |
| Balasubramanian (1997) | | | | | ✓ | | | | ✓ | | | | | |
| Oswald (1983) | | | | | | | | | | | | | | |
| Total | 1 | 0 | 0 | 1 | 9 | 0 | 2 | 0 | 6 | 0 | 1 | 0 | 0 | 0 |

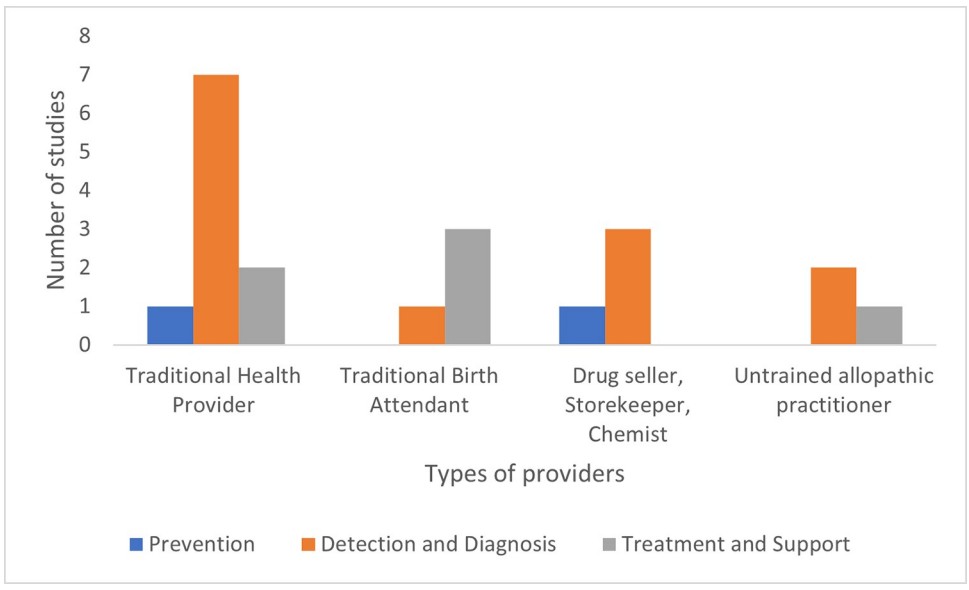

**Fig 3. Bar diagram showing the distribution of IPs role by provider type (n = 21[a]).** [a] Some studies featured more than one type of provider.

Our review has identified that detection and diagnosis (n = 10) was the most prominent role assigned to IPs in TB care. Engaging IPs in this role would address two major challenges faced by TB care programs in LMICs [2]. First, initial care-seeking from IPs by TB patients has been reported in multiple studies [22, 23, 25, 26, 28], and as demonstrated by the studies included in this review [30, 51, 53, 58], including IPs in the TB care would result in timely and

**Table 3. Distribution of outcomes measured in included studies.**

| Study | Outcomes Measured | Domains | Impact of IP's role in TB care outcomes[a] |
|---|---|---|---|
| Sima et al. (2019) | Referrals made | Diagnosis | Positive |
| Dutta et al. (2018) | Case notification | Diagnosis | Positive |
| Bello et al. (2017) | Testing and treatment initiation rate | Diagnosis and treatment | Positive |
| Colvin et al. (2014) | Case notification | Diagnosis | Positive |
| Kaboru et al. (2013) | Referrals made | Diagnosis | Positive |
| Simwaka et al. (2012) | Case notification | Diagnosis | Positive |
| Salim et al. (2006) | Referrals made and treatment outcome | Diagnosis and treatment | Positive |
| Harper et al. (2004) | Referrals made and treatment outcome | Diagnosis and treatment | Positive |
| Colvin et al. (2003) | Treatment outcome | Treatment | Positive |
| Kangangi et al. (2003) | Treatment outcome | Treatment | Positive |
| Jagotal et al. (1997) | Treatment outcome | Treatment | Positive |
| Balasubramanian et al. (1997) | Treatment outcome | Treatment | Positive |
| Oswald et al. (1983) | Referrals made | Diagnosis | Positive |

[a] Positive means that the study has reported improvement in care outcomes like improved referral, increased case notification, higher treatment completion and cure rate.

appropriate referral of TB suspects from the informal to the formal system. A quasi-experimental study by Dutta et al. reports a 30% improvement in TB case notification rate in the intervention arm with the involvement of IPs in case identification [30]. Additionally, Kaboru et al. have noted that assessment of a suspected case by IPs is as effective as when done by a qualified clinician, identifying this as a successful public-private collaboration in TB care [53]. Second, seeking initial care from IPs has been identified as one of the most important factors in diagnostic delay [25, 59, 60]. Therefore, engaging IPs has the potential to promote earlier detection of TB, which is a key TB control and elimination strategy recommended by WHO [61].

Multiple studies included in the review have successfully engaged IPs to administer DOT to patients in the community [32, 50, 54, 56–58]. The positive results observed in these studies are in line with findings from systematic reviews evaluating the impact of similar Community-Based DOT programs [62, 63]. Including IPs as DOT providers capitalises on their flexible working hours, proximity to the community, social and cultural acceptability and trustworthiness [16, 54], all of which can significantly improve TB care outcomes [32, 54, 57]. In Harper et al., all patients successfully completed their six months treatment [32] and similarly, in Jago-tal et al., the treatment adherence was reported to be higher in the group whose DOT was administered by IPs [57]. Colvin et al. reported that satisfaction was higher among patients whose treatment was supervised by IPs, primarily because they enquired about the patient's general health and demonstrated a caring, family-like attitude [55]. Loss to follow up during treatment is an important issue in TB care [64] which may be tackled by engaging IPs. A study conducted in Bangladesh found a treatment success rate of 90%, where IPs were assigned the role of administering DOT [54].

In current literature, there is mixed evidence with regards to IPs treating TB patients. One study in India found that 33.8% of IPs treated TB patients with various drug regimens and all lacked correct knowledge of dose, duration and drug combination [65]. A countrywide cluster randomized survey in Bangladesh reported that IPs prescribed drugs in 80–90% of encounters with patients who visited them up to 4 times with symptoms of TB. However, the study lacks detail on the drugs prescribed by IPs [28]. Similarly, a qualitative study found that IPs treated TB patients for 3–4 months before referral [24]. In this review, we did not find any studies where IPs were assigned the role of initiating treatment for TB patients. The included studies might have only evaluated those roles among IPs that are authorized for non-health professionals as per the WHO International Standards for TB care [66]. The evidence on IPs initiating TB treatment is contested but needs to be a research priority, especially among untrained allopathic practitioners who prescribe antibiotics as part of their practice [67].

There is substantial heterogeneity in background characteristics of IPs by their system of practice (traditional versus modern), including education, experience, training, and service modality, but in the current literature, there exists no classification system for IPs. All IPs are assumed to fit into a single category. This was reflected in the studies included in this review, as none of them reported IPs' characteristics even when studies involved multiple provider types. Hence, future research on IPs should follow the standard protocol of reporting research findings by describing the study participants as this impacts the interpretation and generalizability of findings [68]. Additionally, there is a lack of a clear definition for IPs to enable a uniform approach to research. There is a need to undertake a systematic review of definitions, similar to studies conducted for other community-based health workers [69].

It was evident from this review that quality of care is not yet an important component in research pertaining to IPs, as only two studies in this review measured IPs knowledge on TB care [32, 51]. This finding is similar to a systematic review paper by Sudhinaraset et al., where only 24% of 122 studies reported quality outcomes [16]. Studies have found that IPs lack

appropriate knowledge to deliver health care [21], including TB care [70], and IPs knowledge and skills are identified as a crucial factor that influences the quality of care [16]. Hence, future research should employ validated methods, such as standardized patient methods and vignettes, to measure IPs' knowledge and skills in TB care [71, 72].

Further, we found that none of the included studies tested any kind of digital intervention to improve TB care even though the WHO guideline on the digital tool for END TB Strategy recommends promoting the use of ICT (Information and Communication Technology) [73]. With the increasing recognition of the importance of digital technology in TB care activities [74], replicating and evaluating its use in the informal sector needs to be emphasized.

This review highlights an important role of IPs in TB care, which is often not documented or prioritized in global and national TB reports and guidelines. With many LMICs still struggling to achieve the targets set in the End TB Strategy, it is questionable if these are achievable if the informal sector continues to be ignored. Simply acknowledging that they are the first point of contact for a significant proportion of TB patients is not enough; action needs to be taken, policies need to be expanded, and priorities need to be redefined to accommodate the informal sector in NTPs. It is crucial to address the current global gap with regards to IP engagement as the WHO PPM strategy recommends engaging all identified providers in TB care activities. This review broadly identifies the role of IPs in TB care and presents evidence to highlight the contribution they can make in improving TB care outcomes if appropriate training and support are provided. The feasibility of training IPs to provide quality care is already demonstrated by a randomized control trial conducted in India [19]. It is worth noting that 60% of the studies in this review involved NTPs as collaborators. This indicates that there exists a scope and role for IPs in current NTPs. Engaging IPs will complement and strengthen existing TB care services considering their unique role in the care delivery system, especially in countries with high TB burden. This engagement requires recognition, appreciation, and support from the formal health system to maximize the full potential of a large cadre of the informal health workforce which remains underutilized yet highly valued by their communities in many health systems of LMICs.

## Limitations

We searched for the literature in six of the main databases, but grey literature was not included due to time and scope constraints, and we may have missed some studies, especially the reports of national and international organizations working in the field of IPs. We only included studies published in English. As there is no uniform definition for IPs and various terms are used in different contexts, there is a possibility that we might have missed studies if they have defined IPs outside of our search strategy.

## Conclusions

This review demonstrates the potential of IPs to support prevention, detection, and treatment functions in TB care, leading to improved TB care outcomes. As highlighted by this review, in order to achieve national and global targets, it is essential that NTPs engage with IPs, considering their unique role in TB care. More research is required to answer some of the identified knowledge gaps, and future studies should aim to generate high-quality evidence with methodological rigour. As we progress toward a people-centred model of TB care, and considering the ubiquitous presence of IPs in high burden countries, it is essential to recognise that they can play a valuable role in ensuring acceptable and accessible TB care services are made available to the community. Engaging IPs in TB care would also provide insight on how this large

workforce can be integrated in a health system to strengthen the delivery of primary care, especially in LMICs.

## Supporting information

**S1 File. PRISMA extension for scoping reviews checklist.**
(PDF)

**S2 File. Summary table of IPs definitions in published studies.**
(PDF)

**S3 File. Summary table of WHO People-Centered Model of TB Care.**
(PDF)

**S4 File. Scoping review search strategy in MEDLINE (via Ovid).**
(PDF)

**S5 File. Reasons for exclusion of studies.**
(PDF)

**S6 File. Quality assessment of included studies.**
(PDF)

**S7 File. Classification system for IPs.**
(PDF)

## Acknowledgments

We would like to thank the UNSW library research consultation team for their expertise during the development and refinement of the search strategy.

## Author Contributions

**Conceptualization:** Poshan Thapa, Rohan Jayasuriya, John J. Hall, Padmanesan Narasimhan.

**Data curation:** Poshan Thapa, Nachiket Gudi.

**Formal analysis:** Poshan Thapa, Nachiket Gudi.

**Methodology:** Poshan Thapa, Rohan Jayasuriya, John J. Hall, Kristen Beek, Parthasarathi Mukherjee, Nachiket Gudi, Padmanesan Narasimhan.

**Project administration:** Poshan Thapa.

**Software:** Poshan Thapa.

**Supervision:** Rohan Jayasuriya, John J. Hall, Kristen Beek, Parthasarathi Mukherjee, Padmanesan Narasimhan.

**Visualization:** Poshan Thapa, Nachiket Gudi.

**Writing – original draft:** Poshan Thapa.

**Writing – review & editing:** Poshan Thapa, Rohan Jayasuriya, John J. Hall, Kristen Beek, Parthasarathi Mukherjee, Nachiket Gudi, Padmanesan Narasimhan.

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
