## [Decision Letter · Decision Letter 0]

18 Mar 2021

PONE-D-20-38949

Role of informal healthcare providers in tuberculosis care in low- and middle-income countries: a systematic scoping review

PLOS ONE

Dear Dr. Thapa,

Thank you for submitting your manuscript to PLOS ONE. After careful consideration, we feel that it has merit but does not fully meet PLOS ONE’s publication criteria as it currently stands. Therefore, we invite you to submit a revised version of the manuscript that addresses the points raised during the review process.

Foremost, there are important methodological questions and concerns. The framework for and the accuracy of classification of IP roles is one of these. Similarly, the basis on which studies were excluded, given that some excluded studies were then used in the discussion (and seemingly could have been included given that they provide insight into at least one of the three research questions for the review).

We look forward to receiving your revised manuscript.

Kind regards,

Geoffrey Chan

Academic Editor

PLOS ONE

Journal Requirements:

Reviewers' comments:

Reviewer's Responses to Questions

**Comments to the Author**

1. Is the manuscript technically sound, and do the data support the conclusions?

Reviewer #1: Yes

Reviewer #2: No

Reviewer #3: Yes

2. Has the statistical analysis been performed appropriately and rigorously? 

Reviewer #1: N/A

Reviewer #2: N/A

Reviewer #3: N/A

3. Have the authors made all data underlying the findings in their manuscript fully available?

Reviewer #1: Yes

Reviewer #2: Yes

Reviewer #3: Yes

4. Is the manuscript presented in an intelligible fashion and written in standard English?

Reviewer #1: Yes

Reviewer #2: Yes

Reviewer #3: Yes

5. Review Comments to the Author

Reviewer #1: Well written article. Table 2 "Distribution of outcomes measured in included studies" describes "positive" Direction of Impact, what was the a threshold for this "positive" label? (e.g. p value in quantitative studies). I thought there were several places where the acronym DOTS (the name of the entire five component strategy) was used instead of the more proper DOT (Directly Observed Treatment) for example on line 336, 339 and more. The term "default" was used at least once and should be replaced with the more accepted mouthful "lost to follow-up".

Reviewer #2: This is a well-written study on an important topic that is well framed in the introduction. My concerns are as follows.

1. The proposed definition of Informal Providers (lines 143-145) is somewhat tautological because it twice uses the term “informal” which it purports to define. Many of the definitions found in other publications (file S2) seem more rigorous.

2. IPs are described as “most commonly engaged” or “least engaged” at various points. It would be more accurate to say they “feature most in the studies reviewed” or “feature least in the studies reviewed”, because there is not necessarily a direct relationship between the amount that providers are engaged and the extent to which they feature in published articles.

3. Unqualified Allopathic Practitioners are twice described as “the least engaged provider” (lines 235 and 284). It would be more accurate to say that only one category that features relatively frequently in the reviewed literature (Traditional Healthcare Providers, n=8). The other three categories are about equally rare, each featuring in just 2-3 articles.

4. The desire to align with the “WHO People-Centred Model of TB Care” is understandable, even though the document referenced was designed a “blueprint for eastern European and central Asian countries” whereas those in this review are all in Africa and S Asia. In one study (lines 254-256), the IP role is classified as “prevention”, when “educating people about signs and symptoms of TB and the availability of free services at the local health facility” seems more appropriately classified as “detection” or “case finding”. In 10 studies (lines 257-263) the IP role is classified as “detection and diagnosis”, yet they include very different sub-categories that could usefully be distinguished: 9 studies with passive case-finding, 1 with active case-finding, and 2 in which the IP was tasked with sputum collection rather than just referral of symptomatics. (Figure 3). I would urge you to classify roles in the way that you think would be most useful to your audience. The literature includes many different types of role classifications that may be more useful.

5. Figure 1 reveals that 79 full-text articles were excluded, but the reasons provided are not very informative. The discussion section (lines 350-357) reveals several studies that seem to include very important information about the role of IPs in treating TB that were excluded from the review. Its not clear why they would have been excluded. The implication is that that they may not have undergone ethics approval (line 357) but a quick check of just two of the excluded studies (references 24, 28) reveals that they had ethics approval, so line 357 is misleading and should be deleted. Whatever the criteria used, references 24, 28 and 63 (at least) seem to have been important studies that would have enriched the scoping review; this reviewer did not go through the other 76 studies that were excluded. Given that the stated objective of this scoping review was “to identify the role of IPs in TB care in LMICs” (line 115) the exclusion of so many studies seems to have significantly compromised the objective of the review.

6. Table 2 provides a high-level summary of the outcomes measured and direction of impact for the selected studies. Although this section (and much of the Discussion) goes beyond the stated limited objective of the review, it includes useful information. And even though this was not a systematic review, it would have been useful to summarize also the quality of evidence in each case (since the studies ranged from cross-sectional to cluster-randomized trial), the magnitude of the effect found, and the scale of the interventions or studies.

Reviewer #3: Overall, I thought this was a strong manuscript with clear methodology. I also feel that this is a highly important topic. I commend the authors on their thorough work.

Kindly find my comments below:

Abstract: Kindly clarify what the “gap” is (“However, a gap still exists mainly in engaging…”). Is this a research gap? An implementation gap? An information gap in the guidelines?

In the introduction, I would suggest updating your data to the most recent figures (i.e. the 2020 Global TB report, which has data from 2019).

Line 61: “….Whilst showing a progressive reduction, both fall considerably….” It’s not totally clear what “both” is referring to. Incidence and mortality?

Of note, the term cadre is used several times throughout the paper and is perhaps redundant, unless the authors have reason for using a specific word numerous times (and if so, kindly clarify).

Kindly define “non-qualified” when referring to “non-qualified allopathic practitioners.” I suggest providing a standard definition at the beginning of the manuscript.

Line 97: The consequences of non-engagement with all providers in TB care severely limits the success of NTPs.” While this may be true, we do not actually know this (I would argue that one of the intentions of this review is to demonstrate the value of informal providers). I would change this statement to something along the lines of “potentially limits,” etc.

It is not totally clear to me if you limited the search by time at all (e.g. did you only review papers published between 2000 and 2020?)

Line 332: “Their roles were not different”-I’m not sure what exactly is meant by this

Line 343: I would suggest changing “compliance” to adherence, as this is less stigmatizing/better patient centered terminology

351-352: Kindly clarify what is meant by “various regimens.” Were some incorrect? This is important given the intention of the review.

363: I am not sure what is meant by “sensitive” in this context. I recommend that the authors clarify.

365: I am not sure “undermines” is the right word for this section. I am not actually sure what the authors mean by “this undermines the importance of….system of practice.” I suggest the authors clarify this sentence. In general, I find this paragraph confusing and in need of clarification.

373: “IPs research”-this is a little confusing. Do the authors mean research pertaining to IPs?

Regarding limitations: I encourage the authors to expand slightly on the lack of grey literature; “some studies,” is a bit vague (particularly because grey literature may include policy briefs/reports etc.). I would include a little more information about the choice to not include literature, and what exactly may have been missed, so it is clear why the authors have considered this to be a limitation.

6. PLOS authors have the option to publish the peer review history of their article (what does this mean?). If published, this will include your full peer review and any attached files.

Reviewer #1: No

Reviewer #2: No

Reviewer #3: No

---

## [Author Response · Author response to Decision Letter 0]

26 Jun 2021

26 June 2021

The Editor,

PLOS One

Dear Editor,

Re: Role of informal healthcare providers in tuberculosis care in low- and middle-income countries: a systematic scoping review

We thank the three external reviewers and you for the useful comments that has allowed us to strengthen our manuscript. We have carefully reviewed the manuscript to address the reviewers' and editor's comments. Below, we provide a detailed response to the comments (highlighted in blue). Additionally, we have provided a tracked changed version of the manuscript that highlights the changes made to the originally submitted version.

Academic Editor:

1. Foremost, there are important methodological questions and concerns. The framework for and the accuracy of classification of IP roles is one of these.

Response: We appreciate your feedback. We have carefully considered the reviewer comment and have responded to this in detail under reviewer two comments (serial number 5). 

2. Similarly, the basis on which studies were excluded, given that some excluded studies were then used in the discussion (and seemingly could have been included given that they provide insight into at least one of the three research questions for the review).

Response: Thank you for your comment. These studies referred to in the discussion section (Yellappa, 2017), (Satyanarayana, 2011), (Anandhi, 2002) were not identified during the search process. They were included in the discussion section based on the authors’ knowledge of the subject area. The detailed response to this is included under reviewer two comments (serial number 9 & 11). Additionally, we have included the detailed reason for the exclusion of each study in a supplementary file (S5) in the revised manuscript.

Reviewer 1:

1. Well written article.

Response: We thank the reviewer for appreciating our work. 

2. Table 2 "Distribution of outcomes measured in included studies" describes "positive" Direction of Impact, what was a threshold for this "positive" label? (e.g. p value in quantitative studies).

Response: We thank the reviewer for careful review of this Table. We have made the following changes.

- We have replaced the term "Direction of impact" to "Impact of IP's role in TB care outcomes".

- We have also added a note at the end of Table 3 “Positive means the study has reported improvement in care outcomes like improved referral, increased case notification, higher treatment completion and cure rate”.

Please refer to Table 3 in the revised manuscript.

3. I thought there were several places where the acronym DOTS (the name of the entire five component strategy) was used instead of the more proper DOT (Directly Observed Treatment) for example on line 336, 339 and more.

Response: Thank you for your comment. We have replaced "DOTS" with "DOT" throughout the manuscript in the revised version. For example, please see line 276 in page 10 in the revised manuscript. 

4. The term "default" was used at least once and should be replaced with the more accepted mouthful "lost to follow-up".

Response: We agree with the reviewer and have addressed it in the revised version.

Please see line 346 in page 13 in the revised manuscript.

Reviewer 2:

1. This is a well-written study on an important topic that is well framed in the introduction.

Response: We thank the reviewer for appreciating our work. 

2. The proposed definition of Informal Providers (lines 143-145) is somewhat tautological because it twice uses the term "informal" which it purports to define. Many of the definitions found in other publications (file S2) seem more rigorous.

Response: Thank you for pointing out the tautology. We agree there are a few repetitions of the term informal, so we have made some changes taking into consideration other definitions. 

New version:

We define IPs as "individuals who are not affiliated or registered to any government body or institution, independently delivering some form of health services, and not possessing a recognized certification for the type of services they offer".

As to the rigour of the definition, we have again reviewed the definitions in other publications. There are two that are substantial; Kaboru (2011) and Sudhinaraset (2013). While they contained many of the aspects we selected, we felt a more flexible and broader approach was required considering the nature of the review. There were many considerations in the definition we used in our review, and they were:

1. IPs were not registered or affiliated with a government or a recognized non-governmental organization. We expanded our definition to include private institutions, not limiting it to the government system as in previous studies. Some of the Community Health Workers (CHWs) are not part of a formal system, but they are affiliated to a recognized institution like BRAC NGO in Bangladesh. Restricting it to government affiliation and registration criteria would classify professional CHWs working in NGOs as IPs. 

2. IPs have independently (on their own) established their practice and continue to provide health care in the communities they serve. Examples are traditional birth attendants and village doctors in many societies. They provide health care accepted by the community. It includes traditional and religious healing as well as a modern system of medicine. 

3. Lastly, they do not possess the recognized certification to provide the kind of care they offer in the community, like traditional birth attendants who lack formal midwifery training. 

3. IPs are described as "most commonly engaged" or "least engaged" at various points. It would be more accurate to say they "feature most in the studies reviewed" or "feature least in the studies reviewed", because there is not necessarily a direct relationship between the amount that providers are engaged and the extent to which they feature in published articles.

Response: Thank you for your advice. We have made the changes throughout the manuscript in the revised version as per your suggestion. For example, please see line 288 in page 11 in the revised manuscript. 

4. Unqualified Allopathic Practitioners are twice described as "the least engaged provider" (lines 235 and 284). It would be more accurate to say that only one category that features relatively frequently in the reviewed literature (Traditional Healthcare Providers, n=8). The other three categories are about equally rare, each featuring in just 2-3 articles.

Response: We thank the reviewer for careful review of this section. We have followed your suggestion to replace the term "engage" with "feature", and it seems to flow better in the paper. The changes are incorporated throughout the revised manuscript.

5. The desire to align with the "WHO People-Centred Model of TB Care" is understandable, even though the document referenced was designed a "blueprint for eastern European and central Asian countries" whereas those in this review are all in Africa and S Asia.

Response: We appreciate your advice. In the beginning, before adopting this model, we reviewed the literature to understand the appropriateness of this framework in our review. In global TB guidelines like WHO International Standard of TB care and The End TB strategy, they have described major TB functions but do not have a classification system that could be utilized in our review.

We found the following listed frameworks which provide some kind of classification of TB functions:

Brief guide on tuberculosis control for primary health care providers: https://www.euro.who.int/__data/assets/pdf_file/0015/123162/E82858.pdf

Implementing the stop TB strategy: https://www.who.int/tb/publications/2008/who_htm_tb_2008_401_eng.pdf

However, we selected the WHO People-Centered Model of TB Care, as it was clear, comprehensive, and best suited for the scope of our review. In fact, when analyzing the data, we found that this model was broadly able to capture all the functions we identified in our review. This could be due to the reason that there is a common understanding of TB care functions globally (broadly defined as prevention, detection, and treatment), even though all the services are not available in all parts of the world. Considering these points, we believe that this model is suitable for this review even though it was not designed as a global tool.

We have revised the sentence as below to provide justification for choosing this model in our review.

New version: We defined role based on the functions performed by IPs in TB care. We have used the WHO People-Centered Model of TB Care to classify functions. It provides a granular classification of TB functions and is best suited for this review to comprehensively outline the distinct role of IPs in TB care (43). 

6. In one study (lines 254-256), the IP role is classified as "prevention", when "educating people about signs and symptoms of TB and the availability of free services at the local health facility" seems more appropriately classified as "detection" or "case finding".

Response: Our classification is based on the WHO people-centred Model of TB Care, in which they categorize any activity that is related to awareness and social mobilization under health education and promotion. In the same study (Bello, 2017), IPs were also assigned role to collect sputum samples from suspected patients which was classified under detection and diagnosis. Please refer to the supplementary file (S3), which outlines the framework classification system with definitions.

7. In 10 studies (lines 257-263) the IP role is classified as "detection and diagnosis", yet they include very different sub-categories that could usefully be distinguished: 9 studies with passive case-finding, 1 with active case-finding, and 2 in which the IP was tasked with sputum collection rather than just referral of symptomatic. (Figure 3). I would urge you to classify roles in the way that you think would be most useful to your audience. The literature includes many different types of role classifications that may be more useful.

Response: We thank the reviewer for the suggestion. Based on your advice, we have replaced figure 3 with a table that provides a more detailed distinction of IPs’ roles under all three domains of TB care. Please refer to Table 2 (page 11) in the revised manuscript.

8. Figure 1 reveals that 79 full-text articles were excluded, but the reasons provided are not very informative.

Response: We have updated the PRISMA diagram with reasons as listed below:

- Did not meet study population definition

- Conference abstract

- Exploratory/ Perspective study

- News article

- Study protocol

- Focus on other diseases

- Linked publication

Additionally, we have also provided the reason for excluding each study in a tabular format in the supplementary file (S5).

9. The discussion section (lines 350-357) reveals several studies that seem to include very important information about the role of IPs in treating TB that were excluded from the review. It is not clear why they would have been excluded.

Response: Following are the studies which are referred to in the discussion section between (lines 350-357):

1. Knowledge and practice pattern of non-allopathic indigenous medical practitioners regarding tuberculosis in a rural area of India (Anandhi, 2002).

2. Care seeking in tuberculosis: results from a countrywide cluster randomized survey in Bangladesh (Satyanarayana, 2011).

3. Patients' pathways to tuberculosis diagnosis and treatment in a fragmented health system: a qualitative study from a south Indian district (Yellappa, 2017).

The above studies are surveys (local or national level) and were not identified during the search process. Based on the authors’ knowledge of the subject area, findings of those studies were included in the discussion section. Even if these studies were identified during our initial search, they would not have met the study eligibility criteria as they are cross-sectional and national surveys and do not aim to explore the role of IPs in TB care. We have provided a new supplementary file (S5) explaining the reason for the exclusion of each study.

10. The implication is that that they may not have undergone ethics approval (line 357) but a quick check of just two of the excluded studies (references 24, 28) reveals that they had ethics approval, so line 357 is misleading and should be deleted.

Response: We thank the reviewer for the comment. We agree the writing in the original version of the manuscript is confusing. We have revised that whole paragraph (line 350-361) as below.

New version: In this review, we did not find any studies where IPs were assigned the role of initiating treatment for TB patients. The included studies might have only evaluated those roles among IPs that are authorized for non-health professionals as per the WHO International Standards for TB care (66). The evidence on IPs initiating TB treatment is contested but needs to be a research priority, especially among untrained allopathic practitioners who prescribe antibiotics as part of their practice (67). 

11. Whatever the criteria used, references 24, 28 and 63 (at least) seem to have been important studies that would have enriched the scoping review; this reviewer did not go through the other 76 studies that were excluded.

Response: Following are the studies with references 24, 28 and 63:

24. Patients' pathways to tuberculosis diagnosis and treatment in a fragmented health system: a qualitative study from a south Indian district (Yellappa, 2017).

28. Care seeking in tuberculosis: results from a countrywide cluster randomized survey in Bangladesh (Satyanarayana, 2011).

63. Knowledge and practice pattern of non-allopathic indigenous medical practitioners regarding tuberculosis in a rural area of India (Anandhi, 2002).

These studies were not identified during the search process. Even if these studies were identified during our initial search, they would not have met the study eligibility criteria as they are cross-sectional and national surveys and do not aim to explore the role of IPs in TB care.

12. Given that the stated objective of this scoping review was "to identify the role of IPs in TB care in LMICs" (line 115) the exclusion of so many studies seem to have significantly compromised the objective of the review.

Response: We have followed the JBI guideline for scoping reviews thoroughly to ensure that all the qualified studies were included in this review. The screening and extraction process was conducted by two independent researchers, and the third researcher was involved when there were any discrepancies. After receiving the reviewer comment, we re-verified all the studies to ensure none of the eligible studies were excluded from the review. We have provided the detailed reason for the exclusion of each study in a new supplementary file (S5).

13. Table 2 provides a high-level summary of the outcomes measured and direction of impact for the selected studies. Although this section (and much of the Discussion) goes beyond the stated limited objective of the review, it includes useful information. And even though this was not a systematic review, it would have been useful to summarize also the quality of evidence in each case (since the studies ranged from cross-sectional to cluster-randomized trial), the magnitude of the effect found, and the scale of the interventions or studies.

Response: We agree that even though the protocol for Scoping review we followed by JBI does not require an assessment of quality, it adds valuable information in mapping out evidence on this topic. Therefore, in the revised manuscript, we have conducted a quality appraisal of the included studies using JBI quality appraisal tools.

We have added the following in the methods section in the revised manuscript.

Methodological quality assessment: (line 185-192 in page 7)

The goal of quality assessment (QA) is to provide an overview of the methodological rigour of the included studies. The findings from QA did not limit the inclusion of studies in the review. It was done by two researchers (PT, RJ), and any disagreement was resolved through discussion and consensus. We used JBI's standard critical appraisal tools for experimental and quasi-experimental studies (47). Both these tools have a set of questions that help authors determine the methodological rigour of included studies. Each question is scored 'yes', 'no', 'unclear', or 'not applicable', but there is no overall quality score.

The findings of the QA are as follow:

Quality of studies: (line 227-237 in page 8)

Out of 10 QED studies, only four studies scored more than 4 out of the 9 criteria listed in the checklist. The overall quality of the included studies was found to be low, especially for criteria 5, 6 and 9, indicating potential bias during measurement of outcomes pre and post (32, 51-57) and lack of appropriate statistical analysis (32, 51, 53-55, 57, 58). Five studies reported having a control group (Q4), but in 3 studies, the information about the comparison group was not sufficient (54, 55, 57). One randomized cluster study included in the review was of high quality (10/13) (31). As it was a cluster-randomized design, concealment and blinding of participants was not possible in the study. Future studies should employ standard research methods to address these identified methodological gaps. For detailed QA results, please refer to the supplementary file (S6).

And the following sentence has been added under objective 3 findings.

All studies, including those found to provide high-quality evidence (30, 31, 52), have reported a positive role of IPs in improving TB care outcomes. (line 303-305 in page 12)

Also, a supplementary file (S6) has been added reporting the detailed quality assessment of each included study.

Reviewer 3:

1. Overall, I thought this was a strong manuscript with clear methodology. I also feel that this is a highly important topic. I commend the authors on their thorough work.

Response: We thank the reviewer for appreciating our work. 

2. Abstract: Kindly clarify what the "gap" is ("However, a gap still exists mainly in engaging…"). Is this a research gap? An implementation gap? An information gap in the guidelines?

Response: Thank you for your advice. We have revised this sentence.

New version: The importance of strengthening public-private partnership by engaging all identified providers in Tuberculosis (TB) care has long been advocated in global TB policies and strategies. However, Informal Healthcare Providers are not yet prioritized and engaged in National Tuberculosis Programs globally.

3. In the introduction, I would suggest updating your data to the most recent figures (i.e. the 2020 Global TB report, which has data from 2019).

Response: Thank you for your helpful feedback. We have updated data from the latest TB report 2020.

New version: In 2019, there were an estimated 10 million new cases of TB and 1.2 million deaths among HIV negative people, with an additional 208,000 deaths among HIV positive people (3).

4. Line 61: "….Whilst showing a progressive reduction, both fall considerably…." It's not totally clear what "both" is referring to. Incidence and mortality?

Response: We agree with the reviewer that the previous version needs to be revised for more clarity. We have rewritten this section.

New version: Between 2015-2019, the global cumulative reduction in incidence and mortality was 9% and 14%, respectively. Despite having a progressive reduction, countries with a high TB burden were far from achieving the 2020 milestone of reducing incidence by 20% and mortality by 35% (3, 6).

5. Of note, the term cadre is used several times throughout the paper and is perhaps redundant, unless the authors have reason for using a specific word numerous times (and if so, kindly clarify).

Response: There is no specific reason, and we agree with the reviewer comment. The use of this term in the manuscript has been minimized.

6. Kindly define "non-qualified" when referring to "non-qualified allopathic practitioners." I suggest providing a standard definition at the beginning of the manuscript.

Response: We have replaced the term "Unqualified" with "Untrained" as this better reflects the concept that we are trying to present. We appreciate your feedback, and based on that, we have provided definitions for each category in a separate supplementary file (S7).

7. Line 97: The consequences of non-engagement with all providers in TB care severely limits the success of NTPs." While this may be true, we do not actually know this (I would argue that one of the intentions of this review is to demonstrate the value of informal providers). I would change this statement to something along the lines of "potentially limits," etc.

Response: We thank the reviewer for carefully reviewing this section. We have revised it in the new version.

New version: Non-engagement with all providers in TB care can potentially limit the success of NTPs. Consequences can include increased community transmission, delayed diagnosis and treatment, drug resistance, catastrophic health expenditure for patients, and impaired tuberculosis monitoring and evaluation systems (8, 23, 29).

8. It is not totally clear to me if you limited the search by time at all (e.g. did you only review papers published between 2000 and 2020?)

Response: We made no restriction on date considering the scarcity of evidence in this field. This information is included in the method section under "Context and design". Please see line 160 in page 6. 

9. Line 332: "Their roles were not different"-I'm not sure what exactly is meant by this

Response: We have revised this sentence as "Interestingly, there was no variation in role by provider type". Please see line 322 in page 13 in revised manuscript. 

10. Line 343: I would suggest changing "compliance" to adherence, as this is less stigmatizing/better patient centered terminology

Response: Thank you for your comment. We have replaced it with adherence as suggested. Please see line 343 in page 13 in revised manuscript. 

11. 351-352: Kindly clarify what is meant by "various regimens." Were some incorrect? This is important given the intention of the review.

Response: We have rewritten this sentence in the revised manuscript.

New version: In current literature, there is mixed evidence with regards to IPs treating TB patients. One study in India found that 33.8% of IPs treated TB patients with various drug regimens and all lacked correct knowledge of dose, duration, and drug combination (65).

12. 363: I am not sure what is meant by "sensitive" in this context. I recommend that the authors clarify.

Response: We thank the reviewer for the comment. This section has been revised.

New version: In this review, we did not find any studies where IPs were assigned the role of initiating treatment for TB patients. The included studies might have only evaluated those roles among IPs that are authorized for non-health professionals as per the WHO International Standards for TB care (66). The evidence on IPs initiating TB treatment is contested but needs to be a research priority, especially among untrained allopathic practitioners who prescribe antibiotics as part of their practice (67).

13. 365: I am not sure "undermines" is the right word for this section. I am not actually sure what the authors mean by "this undermines the importance of….system of practice." I suggest the authors clarify this sentence. In general, I find this paragraph confusing and in need of clarification.

Response: We have rewritten this section in the revised manuscript.

New version: There is substantial heterogeneity in background characteristics of IPs by their system of practice (traditional versus modern), including education, experience, training, and service modality, but in the current literature, there exists no classification system for IPs. All IPs are assumed to fit into a single category. This was reflected in the studies included in this review, as none of them reported IPs’ characteristics even when studies involved multiple provider types. Hence, future research on IPs should follow the standard protocol of reporting research findings by describing the study participants as this impacts the interpretation and generalizability of findings (68).

14. 373: "IPs research"-this is a little confusing. Do the authors mean research pertaining to IPs?

Response: Thank you for your comment. This sentence has been revised in the new version.

New version: It was evident from this review that quality of care is not yet an important component in research pertaining to IPs, as only two studies in this review measured IPs knowledge on TB care (32, 51).

15. Regarding limitations: I encourage the authors to expand slightly on the lack of grey literature; "some studies," is a bit vague (particularly because grey literature may include policy briefs/reports etc.). I would include a little more information about the choice to not include literature, and what exactly may have been missed, so it is clear why the authors have considered this to be a limitation.

Response: We thank the reviewer for the comment. We have expanded this section.

New version: We searched for the literature in six of the main databases, but grey literature was not included due to time and scope constraints, and we may have missed some studies, especially the reports of national and international organizations working in the field of IPs.

---

## [Decision Letter · Decision Letter 1]

17 Aug 2021

Role of informal healthcare providers in tuberculosis care in low- and middle-income countries: a systematic scoping review

PONE-D-20-38949R1

Dear Dr. Thapa,

We’re pleased to inform you that your manuscript has been judged scientifically suitable for publication and will be formally accepted for publication once it meets all outstanding technical requirements.

Kind regards,

Geoffrey Chan

Academic Editor

PLOS ONE

Additional Editor Comments (optional):

Reviewers' comments:

Reviewer's Responses to Questions

**Comments to the Author**

1. If the authors have adequately addressed your comments raised in a previous round of review and you feel that this manuscript is now acceptable for publication, you may indicate that here to bypass the “Comments to the Author” section, enter your conflict of interest statement in the “Confidential to Editor” section, and submit your "Accept" recommendation.

Reviewer #1: All comments have been addressed

Reviewer #2: All comments have been addressed

Reviewer #3: All comments have been addressed

2. Is the manuscript technically sound, and do the data support the conclusions?

Reviewer #1: Yes

Reviewer #2: Yes

Reviewer #3: Yes

3. Has the statistical analysis been performed appropriately and rigorously? 

Reviewer #1: N/A

Reviewer #2: Yes

Reviewer #3: N/A

4. Have the authors made all data underlying the findings in their manuscript fully available?

Reviewer #1: Yes

Reviewer #2: Yes

Reviewer #3: Yes

5. Is the manuscript presented in an intelligible fashion and written in standard English?

Reviewer #1: Yes

Reviewer #2: Yes

Reviewer #3: Yes

6. Review Comments to the Author

Reviewer #1: I am happy with the paper. Its main limitation to me is the lack of papers which is stated in the article which also notes that most of the papers that made it through screening were recent publications. No further concerns.

Reviewer #2: (No Response)

Reviewer #3: I commend the authors on this excellent paper with sound methodology. Thank you for this important contribution to the field!

7. PLOS authors have the option to publish the peer review history of their article (what does this mean?). If published, this will include your full peer review and any attached files.

Reviewer #1: No

Reviewer #2: No

Reviewer #3: No

---

## [Editor Report · Acceptance letter]

19 Aug 2021

PONE-D-20-38949R1 

Role of informal healthcare providers in tuberculosis care in low- and middle-income countries: a systematic scoping review 

Dear Dr. Thapa:

I'm pleased to inform you that your manuscript has been deemed suitable for publication in PLOS ONE. Congratulations! Your manuscript is now with our production department. 

Kind regards, 

on behalf of

Mr. Geoffrey Chan 

Academic Editor

PLOS ONE